# Mitochondrial Dysfunction in Diabetic Cardiomyopathy: The Possible Therapeutic Roles of Phenolic Acids

**DOI:** 10.3390/ijms21176043

**Published:** 2020-08-22

**Authors:** Fatin Farhana Jubaidi, Satirah Zainalabidin, Vanitha Mariappan, Siti Balkis Budin

**Affiliations:** 1Center for Diagnostic, Therapeutic and Investigative Studies (CODTIS), Faculty of Health Sciences, Universiti Kebangsaan Malaysia, Jalan Raja Muda Abdul Aziz, Kuala Lumpur 50300, Malaysia; fatinfarhanajubaidi@gmail.com; 2Center for Toxicology and Health Risk Studies (CORE), Faculty of Health Sciences, Universiti Kebangsaan Malaysia, Jalan Raja Muda Abdul Aziz, Kuala Lumpur 50300, Malaysia; satirah@ukm.edu.my (S.Z.); vanitha.ma@gmail.com (V.M.)

**Keywords:** apoptosis, diabetes mellitus, natural products, inflammation, oxidative stress

## Abstract

As the powerhouse of the cells, mitochondria play a very important role in ensuring that cells continue to function. Mitochondrial dysfunction is one of the main factors contributing to the development of cardiomyopathy in diabetes mellitus. In early development of diabetic cardiomyopathy (DCM), patients present with myocardial fibrosis, dysfunctional remodeling and diastolic dysfunction, which later develop into systolic dysfunction and eventually heart failure. Cardiac mitochondrial dysfunction has been implicated in the development and progression of DCM. Thus, it is important to develop novel therapeutics in order to prevent the progression of DCM, especially by targeting mitochondrial dysfunction. To date, a number of studies have reported the potential of phenolic acids in exerting the cardioprotective effect by combating mitochondrial dysfunction, implicating its potential to be adopted in DCM therapies. Therefore, the aim of this review is to provide a concise overview of mitochondrial dysfunction in the development of DCM and the potential role of phenolic acids in combating cardiac mitochondrial dysfunction. Such information can be used for future development of phenolic acids as means of treating DCM by alleviating the cardiac mitochondrial dysfunction.

## 1. Introduction

For the past 50 years, our knowledge on the pathophysiology of cardiac failure has advanced significantly. Nevertheless, cardiac diseases (including coronary heart disease and stroke) are the most common non-communicable diseases globally, with an estimated 17.8 million deaths in 2017 (more than three quarters were from low-income and middle-income countries) [1]. Presently, it is also estimated that 451 million people are suffering with diabetes worldwide, and the number is forecasted to climb up to 631 million by year 2045, particularly in low- and middle-income countries [2]. Both type 1 (juvenile-onset or insulin-dependent diabetes, T1DM) and type 2 (adult-onset or non-insulin-dependent diabetes, T2DM) diabetes mellitus patients are at greater risk of developing heart disease. It is notable that patients with diabetes are at higher risk (almost two times) of cardiac diseases associated with mortality than similar patients without diabetes. Interestingly, developing cardiovascular disease is the most harmful consequence of diabetes mellitus, and the risk of cardiovascular complications is doubled in patients with diabetes compared to non-diabetic populations [3].

Rubler et al. [4] described patients with diabetes suffering from heart failure but with normal coronary arteries and no other distinct etiologies apparent for heart failure. Moreover, their finding has been reaffirmed by other researchers in several subsequent studies [5,6]. These observations have led to the coining of the new “diabetic cardiomyopathy” (DCM), which elucidates ventricular dysfunction occurring in patients with diabetes (not attributed primarily to coronary artery disease and hypertension). The majority of the patients with T2DM, which is closely associated with obesity and a prolonged sedentary lifestyle. Cases of T2DM being diagnosed in patients under 30 years old are increasingly reported, although T2DM used to be a rare occurrence in young people [7,8]. In chronic models of T1DM mice, the dysregulation in mitochondrial biogenesis is associated with impairment in mitochondrial function and alteration in ultrastructure [9]. In addition, a study using genetically T2DM mice (ob/ob) showed them to be defective in mitochondrial function, which is associated with a reduction in mitochondrial oxidative capacity, increased mitochondrial uncoupling and reduced ATP production, contributing to cardiac dysfunction [10]. A study using mice with cardiomyocyte deletion of insulin receptors (CIRKO) showed that myocardial insulin signaling impairment promotes oxidative stress and mitochondrial uncoupling, which then alters mitochondrial energetics.

While there are limited studies conducted in relation to cardiac mitochondrial function in diabetic patients, there is some indirect evidence to relate the diabetic condition to mitochondrial dysfunction. In well-controlled patients with T2DM, it was found that there was an association between diastolic dysfunction and the reduction in cardiac phosphocreatine/ATP ratios [11]. Although the pathophysiological mechanisms leading to the increased risk for the development of heart failure are certainly multi-factorial, more rising evidence suggests that disarrangements in cardiac energy metabolism play a fundamental role in the pathogenesis of DCM.

Over the years, several researchers have addressed their concern regarding cardiovascular dysfunction in patients with diabetes in many aspects related to mitochondrial dysfunction. Bugger and Abel [12] have discussed several potential mechanisms that may contribute to changes in myocardial mitochondrial energetics in type 1 and type 2 diabetic hearts. However, they suggested the likelihood that not one single mechanism, but rather the combination of several mechanisms, may lead to mitochondrial dysfunction. In addition, Nunes et al. [13] reviewed the improved knowledge regarding the molecular mechanisms underlying DCM development, which contributed to the identification of novel putative targets and therapeutic opportunities for the management of DCM. Based on primary DCM pathophysiological events, Jia et al. [14] substantially reviewed the role of pathogenesis and the role of the molecular proteins and signaling pathways in hyperglycemic and insulin-resistant DCM.

Mitochondria serve as the powerhouses of a cell, and recent reports implicate mitochondrial injury to be a major player in the pathophysiology of diabetic heart disease [15]. Therefore, strategies to attenuate mitochondrial injury might be a potential therapeutic target for diabetic heart disease. Here, we discussed the potential role of phenolic acids (ferulic, ellagic, gallic, salvianolic, chlorogenic, rosmarinic, vanillic and caffeic acids) in combating cardiac mitochondrial dysfunction in order to comprehend the use of these phenolic acids as targets for the development of therapeutic strategies that may prevent DCM. The literature search was completed using the PubMed database and relevant keywords (mitochondrial dysfunction; diabetic cardiomyopathy; in vitro; in vivo; inflammation; apoptosis; oxidative stress; fibrosis; hypertrophy) were used as a search strategy. Based on the search findings, we summarized the articles and integrated the literature search accordingly. In addition to phenolic acid, we included: ferulic acid or ellagic acid or gallic acid or salvianolic acid or chlorogenic acid or rosmarinic acid or vanillic acid or caffeic acid. These phenolic acids were chosen based on previous research, as they are widely found in natural products and have been proven to have therapeutic effects, especially on cardiac diseases.

## 2. Diabetic Cardiomyopathy

Cardiomyopathies are a heterogeneous group of diseases of the myocardium associated with mechanical and/or electrical dysfunction that usually exhibit inappropriate ventricular hypertrophy or dilatation. Cardiomyopathies are either confined to the heart or are part of generalized systemic disorders, often leading to cardiovascular death or progressive heart failure-related disability [16]. In that regard, DCM is defined as the presence of abnormal myocardial function and structure while in the absence of other heart disease risk factors, including coronary artery disease, hypertension and valvular disease, in patients with diabetes mellitus [17]. DCM is typically presented with a marked increase in low density lipoprotein (LDL), glucose, glycated hemoglobin (HbA1c) levels, fibrotic markers insulin-like growth factor (IGF)-B7 and transforming growth factor (TGF)-β1 levels, along with severe diastolic dysfunction [18].

DCM is initially described by myocardial fibrosis, dysfunctional remodeling, and associated diastolic dysfunction, soon after by systolic dysfunction, and ultimately by clinical heart failure. Several mechanisms have been implicated in the development and progression of DCM, including disruption in cardiac insulin signaling, mitochondrial dysfunction, oxidative stress, inflammation, impaired calcium handling, endoplasmic reticulum stress, microvascular dysfunction, renin–angiotensin–aldosterone system (RAAS) activation, and a multitude of cardiac metabolic abnormalities [19,20,21,22,23,24,25,26]. Among all complications of diabetes, DCM contributes to nearly 80% of mortality in patients with diabetes mellitus [27].

## 3. Structural and Functional Alterations in DCM

As mentioned earlier, Rubler et al. [4] was among the earliest reports on structural changes that occurred in diabetic myocardium. In the absence of hypertension, coronary artery disease or valvular heart disease, Rubler and colleagues discovered heart failure-related attributes on excised diabetic hearts, namely, left ventricular hypertrophy and myocardial fibrosis. Generally assessed by non-invasive techniques such as echocardiography or magnetic resonance imaging (MRI), DCM is characterized by the hallmark myocardial hypertrophy and fibrosis [28,29,30]. Both echocardiography and MRI are equally sensitive in detecting structural abnormalities on the heart [29]. However, MRI has the capability to detect more pathological abnormalities. Considering the cost of running an MRI and the fact that it is not as widely available as echocardiography, with comparable sensitivity, echocardiography is still a reliable diagnostic tool for early detection of DCM [31].

Fibrosis plays an important role in structural alterations of the heart in DCM [14]. The myocardial fibrosis event, which contributes majorly to the impaired left ventricular function, is observed in DCM patients. Diastolic dysfunction is one of the earliest signs of DCM and may be detected by imaging techniques. It is characterized by abnormal relaxation and filling of the heart, typically manifested by reduced early diastolic filling and increased atrial filling, increased isovolumetric relaxation time and supraventricular premature beats, as well as by amplified left ventricular end-diastolic pressure and reduced left ventricular end-diastolic volume [13,32,33]. Although diastolic dysfunction is also a feature of other cardiovascular diseases such as coronary artery disease, hypertension and hypertrophic cardiomyopathy, the absence of congestive heart failure manifestation may also represent early DCM development [34]. Systolic dysfunction or the impaired ability of the heart to pump arterial blood is associated with the reduction in the left ventricular ejection fraction and cardiac output. In contrast to diastolic dysfunction, in DCM, systolic dysfunction occurs in the later, usually when patients have already developed deleterious diastolic dysfunction [12]. Brief summary of molecular pathogenesis of structural and functional alterations in DCM is illustrated in Figure 1.

In DCM, fibrosis is commonly observed in diabetic hearts whereby increased perivascular and intermyofibrillar fibrosis have been observed previously in the absence of coronary artery disease and hypertension [35,36]. Fibrosis along with the structural and functional changes was observed in the heart of diabetic rat model [37]. Increased expression of TGF-β1 and the promoter of collagen production, connective tissue growth factor (CTGF), may contribute to the increased collagen deposition in the diabetic heart. Besides, increased activation of poly (ADP-ribose) polymerase 1 (PARP-1) might also play a role [38]. Remodeling of matrix metalloproteinases (MMPs) also causes dysregulation of the extracellular matrix of cardiomyocytes, thus increasing connective tissue content in diabetic hearts [39]. Since adult cardiomyocytes rarely proliferate, the loss of cardiomyocytes eventually leads to compromised cardiac function.

The formation of myocardial fibrosis in DCM involves stiff collagen deposition and its cross-linking, interstitial fibrosis, thickening and sclerosis of small coronary vessels, thickening of the basal membrane, as well as microvascular sclerosis and coronary microaneurysm [13]. Activation of RAAS and the sympathetic nerve system, stimulation of AGE-mediated signaling on the cell surface, hyperinsulinemia and hyperglycemia together cause the activation of the TGF-β1 pathway and uncontrollable extracellular matrix degradation. Additionally, these events also promote myocyte necrosis and apoptosis in DCM [40,41]. Myocyte necrosis results in the extracellular compartment widening and deposition of collagen, consequently causing fibrosis and connective tissue proliferation [42].

Myocardial hypertrophy results from apoptosis and necrosis events of cardiomyocytes, leaving the viable cardiomyocyte to enlarge as a compensation response. Left ventricular hypertrophy is more common in DCM and is a sign of late development of the disease. Prolonged hyperinsulinemia is found to cause increased myocardial mass and decreased cardiac output [30,43]. Moreover, over-expression of hypertrophic genes, such as β-myosin heavy chain (β-MHC), atrial natriuretic peptide (ANP) and brain natriuretic peptide (BNP), represents cardiomyocyte hypertrophy, along with altered width and disorganization of cardiomyocyte myofiber [44,45].

Beside the development of fibrosis and hypertrophy of the heart, pathological changes in the cardiac molecular events could lead to cardiomyocyte stiffness that later develop into diastolic dysfunction. Mechanisms that lead to cardiomyocyte stiffness of diabetic heart includes incomplete metabolic insulin signaling, which reduces the recruitment of glucose transporter type 4 (GLUT4) to plasma membrane, and glucose intake, which reduces sarcoplasmic reticulum calcium ion (Ca^2+^) pumps and increases the level of Ca^2+^ in the intracellular of cardiomyocytes [22,26,46]. Meanwhile, abnormal metabolic insulin signaling also reduces the activity of insulin-stimulated coronary endothelial nitric oxide synthase (eNOS), which in return increases cardiomyocyte sensitization to Ca^2+^ and reduces sarcoplasmic Ca^2+^ intake [47]. Reduction in nitric oxide bioavailability results in titin phosphorylation. This pathophysiological abnormality promotes cardiac stiffness which disturbs its relaxation, leading to diastolic dysfunction, i.e., the primary manifestation of DCM [48].

## 4. Cardiac Mitochondria in Diabetic Cardiomyopathy

Mitochondria are semi-autonomous organelles in a cell and are encapsulated in a double membrane structure. Their size varies from 0.75 to 8 µm [34]. Mitochondria play a pivotal role in cellular energy transduction. Therefore, any disturbance that alters mitochondria efficiency eventually induces excessive production of oxidative stress, which promotes the development of cardiovascular diseases [49]. The heart, an organ with high demand of aerobic metabolism, is greatly affected by mitochondrial dysfunction [50]. One-third of the heart’s volume is composed of mitochondria alone as they are essential organelles in the production of adenosine triphosphate (ATP) from the oxidation of fatty acid and glucose [51].

Electrons from nicotinamide adenine dinucleotide (NADH) and flavin adenine dinucleotide (FADH2) oxidations are funneled through the electron transport chain in the inner membrane of the mitochondria [52]. Accompanied by the translocation of protons across the inner mitochondrial membrane to the intermembrane space, an electrochemical gradient is generated [53]. ATP is generated from the energy of the electrochemical gradient via the ATP synthetic machinery, which is driven by the collapse of a proton gradient through the ATP synthase [54]. However, when the electrochemical potential difference generated by the electrochemical gradient is high (such as in high-fat or high-glucose states, or under conditions of inhibition of electron transport chain (ETC) complexes), the life of superoxide-generating electron transport intermediates, thus, the ubisemiquinone radical is prolonged, resulting in increased production of reactive oxygen species (ROS) [55].

As the heart demands a high amount of energy to operate, the development of DCM has been frequently associated with mitochondrial dysfunction. Reductions in ATP levels are found to be associated with left ventricular cardiac hypertrophy and suppressed systolic function as observed via nuclear magnetic resonance (NMR) imaging [56], which suggests correlation of DCM with mitochondrial dysfunction.

## 5. Involvement of Mitochondria Dysfunction in Diabetic Cardiomyopathy

Mitochondrial dysfunction plays a significant part in DCM development [57]. Around 90% of ATP produced by cardiomyocytes is produced via oxidative phosphorylation in the mitochondria. In the diabetic condition, due to the lack of insulin or its action, mitochondria will switch the source of ATP production from glucose to fatty acid oxidation in order to sustain sufficient ATP production. This process generates more ROS and disrupts the oxidative phosphorylation process [58]. As a result, cell death occurs following impairment of mitochondrial Ca^2+^ handling [59]. Mitochondrial respiratory dysfunction would then follow. Cytosolic Ca^2+^ overload, triggered by excessive calcium influx or reduced calcium efflux, may induce the opening of mitochondrial permeability transition pore (mPTP), leading to mitochondrial dysfunction. [60]. Figure 2 briefly summarize the role of mitochondrial dysfunction in DCM.

### 5.1. Oxidative Stress

Oxidative stress, an imbalance in reactive ROS and antioxidant levels, plays an important role in the development and progression of diabetes and its complications [61]. ROS exerts damage onto proteins and lipid components of cells via oxidation, oxidizing lipids into reactive lipid peroxides [62]. Mitochondrial ROS have also been related to the increased activity of uncoupling proteins (UCP), which uncouple ATP synthesis from electron transport. UCP activity leads to heat generation without ATP production, and long-term reductions in ATP levels affect cellular insulin signaling [63]. Increased oxidative stress is a key contributor to hyperglycemia-induced damage. In DCM, hyperglycemia induces ROS production via the activation of NADPH oxidases, xanthine oxidase and NO synthase. This in turn generates superoxides, which then undergo rapid dismutation to hydrogen peroxide (H_2_O_2_) [64]. Deoxyribonucleic acid (DNA) is especially susceptible to ROS-induced damage, including mitochondrial DNA [65].

DCM and heart failure development and progression worsen by the generation of oxidative stress [66]. Mitochondrial ROS elicits cellular damage via activation of multiple pathways in hyperglycemia [67]. In the mitochondrial ETC process, oxygen metabolism at complex I and III produces ROS. ATP is synthesized via the electrochemical proton gradient in the mitochondria in the normal condition [68]. However, increased NADH and FADH2 flux to the mitochondrial chain promoted by hyperglycemia and insulin resistance results in the mitochondrial inner membrane’s hyperpolarization, complex III inhibition and excessive ROS production [69]. Nicotinamide adenine dinucleotide phosphate (NADPH) oxidase, a membrane-bound enzyme complex, is known to be an important source of cardiomyocyte ROS, and its level increases in the diabetic condition [70]. Activation of the profibrotic transforming growth factor β1/Smad 2/3 signaling pathway by increased RAAS-mediated NADPH oxidase activity may also directly promote cardiac fibrosis, leading to DCM [71,72].

Chen and colleagues [73] demonstrated that over-expression of catalase or manganese superoxide dismutase in the diabetic heart via the transgenic method was able to restore impaired mitochondrial function and cardiomyocyte contractility, showing that ROS does play a role in the pathogenesis of DCM. The inhibition of oxidative phosphorylation (OXPHOS) complex I and III and over-expression of catalase further confirmed that these ROS were generated by mitochondria [74]. This shows that mitochondria in diabetic hearts overproduce ROS, which is made worse by hyperglycemic conditions. Excessive generation of ROS in diabetic myocardium results in myocardial damage at molecular and cellular levels, which, in turn, causes cardiac morphological and functional abnormalities [75].

In the diabetic heart, oxidative stress, following imbalance of antioxidant defenses and accumulation of oxidants, makes mitochondria susceptible to oxidative damage [76]. Consequently, this leads to mitochondrial DNA and lipid damage and inactivation of the ETC complexes and mitochondrial proteins, thus impairing OXPHOS and promoting ROS accumulation in the cells [45]. In addition, cytosolic ROS production may play a role in inducing cardiac mitochondrial damage and even enhance mitochondrial ROS production. Cytosolic ROS are known to induce the opening of the mPTP [77]. As high levels of AGE are typically found in diabetic patients, the production of AGEs associated with hyperglycemia—which results in AGE receptor (RAGE)-induced production of cytosolic ROS—consequently results in the opening of the mPTP [78].

Mitochondrial DNA is extremely sensitive to oxidative damage due to its proximity to the inner membrane, insufficient repair mechanisms and lack of protective histones [79]. Culminating mutations of DNA resulting from oxidative damage leads to further imbalanced ETC and increased ROS production, thus maintaining oxidative stress and enhancing myocardial hypertrophy, fibrosis and cell death, as well as inflammation in the diabetic condition [80]. Therefore, manipulating the mechanisms that are able to eliminate dysfunctional mitochondria is essential to prevent oxidative damage exerted by ROS and, by extension, the development of DCM.

### 5.2. Inflammation

Inflammation is well promoted in the diabetic state, whereby the presence of an elevated concentration of C-reactive proteins (CRP) and other inflammatory markers in the cardiac tissues of diabetic animals suggests the roles of inflammation in the development of DCM [81]. Intramyocardial inflammation in DCM is demonstrated by a marked increased expression of cell adhesion molecules (ICAM-1) and vascular cell adhesion molecule 1 (VCAM-1), increased infiltration of macrophages and leukocytes and increased expression of inflammatory cytokines (interleukin (IL)-1β, IL-6, IL-18, tumor necrosis factor α (TNFα) and TGF-β1) [82]. There is a strong link between oxidative stress and inflammation in T2DM that mediates the development of various complications, including DCM [83]. Oxidative stress can be provoked by proinflammatory cytokines during the removal of pathogens via the generation of ROS [84]. Chronic inflammation is known as a prolonged pathological condition that could lead to ROS overproduction from the activity of inflammatory cells, consequently causing cell damage, tissue destruction and fibrosis [73].

In DCM, mitochondrial dysfunction and systemic oxidative stress lead to the destruction of proteins, lipids and nucleic acid. This results in cardiomyocyte death as well as propagation of inflammatory cytokines, as seen in both animal models and human studies [85,86]. In regard to mitochondrial dysfunction-derived inflammation, the recruitment and assembly of an inflammatory complex known as inflammasome plays a key role [87]. Diabetes-derived systemic tissue damage leads to the activation of Toll-like receptors (TLR), especially TLR2 and TLR4 on the macrophages, which in turn induce mitochondrial damage and lysosomal stress [88,89]. Release of mitochondrial DNA and ROS may activate the nucleotide oligomerization domain (NOD)-like receptor containing pyrin domain 3 (NLRP3). NLRP3 is a key mediator in activating the innate immune system in response to various signals. Upon activation, NLRP3 triggers the assembly of inflammasome by self-oligomerization, leading to the cleavage of proinflammatory cytokines IL-1β and IL-18 via activation of the caspase-1 mechanism [90]. It has been previously reported that NLRP3 inflammasome is hyperactivated in the diabetic heart, thus promoting cardiac inflammation [91]. The release of oxidized mitochondrial DNA into the cytosol as well as lysosomal stress were also found to promote inflammasome assembly [91,92]. The elaboration of the proinflammatory cytokines further generates oxidative stress and mitochondrial dysfunction, thus contributing to the development of DCM.

### 5.3. Apoptosis

Programmed cell death or apoptosis is the fastest form of cell death. Increase in apoptotic and necrotic cell death is often observed in diabetic hearts [93,94]. Although necrosis is the main cause of cardiomyocyte loss, previous studies suggested that apoptosis could also occur following necrosis events that lead to cardiac structural changes and functional loss [95]. Apoptosis of cardiomyocytes can be initiated by mitochondrial dysfunction in the presence of impaired activity of the ETC, ATP depletion and increased oxidative stress [96]. Cytochrome C forms a multi-protein complex with apoptotic protease-activating factor 1 (APAF1), which then activates procaspase-9. Once activated, this initiator caspase proteolytically activates the effectors caspases and triggers a cascade of irreversible events leading to apoptosis [97].

The increase in cardiac apoptosis is a major event in the development of DCM. Compared with healthy individuals, patients with diabetes mellitus were characterized by an 85-fold increase in cardiomyocytes apoptosis with only 4-fold increases in cardiomyocyte necrosis. Thus, this suggests that cardiomyocytes are more susceptible to diabetes-induced cell death by apoptosis than by necrosis [98,99]. Cardiomyocyte apoptosis leads to cell loss, to decreased cardiac contractile function and ultimately to the promotion of cardiac remodeling [100].

Activation of proapoptotic signaling pathways as well as necrosis may also be activated by mechanisms that increase fibrosis and myocardial inflammation. Mechanisms responsible for the increased rates of apoptosis include increased production of ROS, endoplasmic reticulum stress, increased activation of the TGF-β1 signaling pathway, increased circulating inflammatory cytokines and chemokines, increased RAAS local activation, increased caspase activation, resistance to IGF-1 and altered expression of pro- and anti-apoptotic molecules [101,102,103,104,105,106,107]. Mitochondrial oxidative stress was found to be alleviated by metallothionein treatment in streptozotocin-treated mice, attenuating cell death [106]. Increased expression of glucose-regulated protein 78 (Grp78) and caspase12 in diabetic heart indicates the activation of endoplasmic reticulum-stress response that leads to cell death via apoptosis. [105].

Glycogen synthase kinase-3 β (GSK-3 β) induces apoptosis via activation of the apoptosis-related protease-caspase family and B-cell lymphoma 2 associated x-protein (BAX) genes [106]. Xu et al. [107] reported that metformin was able to restore autophagic activity and inhibit cardiomyocyte apoptosis via disruption of B-cell lymphoma 2 (Bcl-2) and Beclin-1 complex in diabetic heart tissue. Activation of the autophagic response is considered a compensatory feedback mechanism in protecting against cell apoptosis and maintaining normal cellular function in the diabetic condition [107].

Increased caspase activity in diabetic mice was associated with perturbation in mitochondrial membrane potential and the opening of the mPTP [108]. Alterations in the expression of mitochondrial adenine nucleotide translocase 1 (ANT1) and antioxidant, which both are contributors to mitochondrial-induced cell death, were also observed in diabetic mice. Mitochondrial protein and cardiolipin were also reported to undergo oxidative damage, which may perturb mitochondrial membrane integrity, thus inducing mitochondrial apoptosis [109,110]. Significant loss of cardiomyocyte follows if cardiomyocyte apoptosis events are not being inhibited, eventually resulting in cardiac fibrosis. Therefore, the inhibition of cardiac apoptosis is an important strategy for the prevention of diabetic cardiomyopathy.

### 5.4. Reduced Mitochondrial ATP Production

Mitochondria synthesize cellular ATP via the OXPHOS mechanism in the electron transport chain [111]. As the heart uses a huge amount of ATP due to high energy consumption, it contains a relatively low ATP reservoir. Respiratory chain activity is controlled by mitochondrial creatine kinase, integrating high-energy phosphate metabolism [112]. Phosphocreatine plays an important role in maintaining ATP buffer content in the cardiac muscle and is produced by the creatinine in the creatinine kinase enzyme. There are two forms of phosphocreatine: the highly active octameric form and the less active dimeric form. Both forms exist in a dynamic balance. However, in the pathologic condition, the concentration of the dimeric form will be higher because of the dissociation of the octameric form [113]. This results in the impairment of the respiratory control and compensation of high cardiac ATP consumption by the hydrolysis of the phosphocreatine. Impairment of OXPHOS and low ATP production lead to the reduction in cardiac performance in a diseased heart [114].

Mitochondrial oxidation of fatty acid provides 50–70% of energy needed by the adult human heart [115]. Myocardium is capable of switching the source of energy from lipid to glucose depending on the nutrient availability in order to maintain stable ATP production. Insulin-dependent signaling plays a role in the regulation of the selection of source of energy [116]. However, as insulin signaling is down-regulated or perturbed in the diabetic condition, this metabolic flexibility is lost. Consequently, the cardiac ATP reservoir is depleted as the reduction in fatty acid oxidation is not re-sourced to glucose oxidation [117].

### 5.5. Insulin Resistance

Insulin resistance is commonly associated with T2DM, as this type of diabetes mellitus arises from impaired insulin action, thus impeding insulin signaling. However, studies have shown that insulin resistance may developed in T1DM patients too [118]. This condition is termed “double diabetes”, whereby hyperglycemia is present in children and young adolescents with the combination of clinical features and markers typically found in both T1DM and T2DM [119]. Insulin resistance is also associated with diminished mitochondrial density, alterations in mitochondrial structure, increased mitochondrial oxidative stress and reduced production of ATP [120].

Reduction in mitochondrial biogenesis and content as well as impairment of the ETC process result in decreased substrate oxidation in the mitochondria. Due to the absence of insulin and insulin resistance, cardiomyocytes undergo a metabolic shift to β-oxidation in order to maintain ATP production. However, this process would eventually unable to metabolize all incoming fatty acids and results in intracellular lipid accumulation [121]. Deposition of lipid diacylglycerols (DAG) activates protein kinase, translocating to the plasma membrane and inhibiting insulin receptors [122]. Deposition of lipid ceramide (CER) also alters insulin signaling via inhibition of protein kinase B (Akt) [123]. Aside from lipid accumulation, a decrease in substrate oxidation may also lead to electron leakage due to impairment of electron flow in ETC, promoting the formation of superoxides from the free oxygen molecules. Consequently, this leads to the generation of oxidative stress, which induces lipid, protein and nucleic acid damage of cardiomyocytes. Removal of mitochondria via mitophagy further reduces the number of viable mitochondria, resulting in decreased substrate oxidation and further worsening lipid accumulation [124]. Due to the fact that the heart consumes huge amounts of energy, a small disruption in substrate oxidation may potentially lead to lipid accumulation and subsequently insulin resistance [26].

Incomplete or disrupted insulin signaling contributes to cardiomyopathy via two key pathways: insulin receptor substrate-1 (IRS-1) and mitogen-activated protein kinase (MAPK). IRS-1 elicits a metabolic response by acting on the upstream of the P13K-Akt signal transduction pathway. This initiates a kinase cascade that further activates the Akt signaling pathway, promoting glucose uptake by allowing GLUT4 to translocate into the cell membrane [125]. MAPK, on the other hand, plays a role in the growth and remodeling responses of the heart, of which activation may lead to myocardial hypertrophy, fibrosis or impaired myocardial–endothelial signaling [126].

## 6. Phenolic Acids: Therapeutic Implications for Mitochondrial Dysfunction

Phenolic acids can be divided into two major groups: hydroxybenzoic acids and hydroxycinnamic acids. Hydroxybenzoic acids are derived from non-phenolic molecules of benzoic acid, whereas hydroxycinnamic acids are from non-phenolic molecules of cinnamic acid [127]. The term phenolic acid describes a phenol ring that possesses at least one carboxylic acid functionality. Phenolic acids are usually sub-classified into benzoic acids containing seven carbon atoms (C6–C1) and cinnamic acids with nine carbon atoms (C6–C3) [128].

Phenolic acids are widely distributed in the plant kingdom and found in a wide variety of nuts and fruits, such as raspberries, grapes, strawberries, walnuts, cranberries, and blackcurrants [127]. These are secondary metabolites derived from phenylalanine and tyrosine via the shikimate/chorismate pathway. However, these compounds exist predominantly as hydroxybenzoic acids (including gallic acid, salicylic acid, protocatechuic acid, ellagic acid, and gentisic acid) and hydroxycinnamic acids (including include p-coumaric acid, caffeic acid, ferulic acid, chlorogenic acid, and sinapic acids) or, alternatively, they may occur as conjugated forms. Growing evidence indicates that various dietary phenolic acid may prevent cardiac mitochondrial dysfunction. Table 1 summarizes the therapeutic potential of phenolic acids on cardiac mitochondrial dysfunction.

### 6.1. Ferulic Acid

Ferulic acid is a phenolic compound that is clinically used to treat angina pectoris and hypertensive diseases [129]. Ferulic acid acts by scavenging free radicals and quenching lipid peroxidative chains. The hydroxy and phenoxy groups present in ferulic acid donate their electrons to the free radicals, reducing their capacity to damage [130]. The potent antioxidant property of ferulic acid is contributed by its phenolic nucleus and unsaturated side chain that readily forms a resonance-stabilized phenoxy radical [131].

Perez-Ternero et al. [132] reported that ferulic acid was able to improve expression of mitochondrial biogenesis and dynamic markers, as well as reduce oxidative stress in mouse models. Treatment of ferulic acid significantly alleviates the disturbance in mitochondrial activity and ATP levels, restoring the cardiac mitochondrial function of isoproterenol-induced rats [130]. Ferulic acid is found to be able to prevent mitochondrial dysfunction induced by the hyperglycemia condition in H9c2 cells via the sarcoplasmic reticulum Ca^2+^-ATPase (SERCA)/phospholamban (PLN) pathway [133]. SERCA/PLN plays a very significant role in transporting Ca^2+^ into the sarcoplasmic reticulum as well as in regulating Ca^2+^ homeostasis of the cells. In the hyperglycemic condition, down-regulation of the SERCA/PLN pathway leads to a decrease in sarcoplasmic reticulum Ca^2+^ transport, causing significant intracellular Ca^2+^ overload [134,135].

Song et al. [136] suggested that ferulic acid mitigated oxidative damage via antioxidative protection by heme oxygenase-1 (HO-1) up-regulation. Ferulic acid induces HO-1 up-regulation through the Kelch-like ECH-associated protein 1-Nrf2-antioxidant response element (Kaep1-Nrf2-ARE) pathway. As a key regulator of HO-1, Nrf2 binds to ARE, which is a *cis*-acting enhancing sequence that mediates the transcriptional activation of Nrf2 in response to oxidative stress [137]. In the normal condition, Nrf2 is an inactive complex, and its repressor, Kaep1, negatively regulates Nrf2 by ubiquitination and proteosomal degradation [138]. When under oxidative stress, Nrf2 dissociates from Kaep1, translocates into the nucleus and binds to ARE. This results in the regulation of genes such as HO-1 and glutathione transferase [139].

Moreover, Song et al. [140] also found that hyperglycemia-induced AGE fluctuation was significantly alleviated upon ferulic acid treatment. This suggests that ferulic acid can prevent hyperglycemia-induced cell viability, apoptosis, down-regulation of Nrf2 protein and up-regulation of Kaep1 protein expression in cardiomyocytes [137,141].

### 6.2. Ellagic Acid

Ellagic acid is commonly found in fruits, vegetables and berries. Ellagic acid was reported to promote cell cycle exit in cancer cells, inhibiting them from proliferation. In addition, ellagic acid also demonstrated antioxidative properties in various pathological conditions, such as nephrotoxicity and hypertension [142,143].

A previous study found that ellagic acid was able to suppress oxidative injury on mitochondria induced by both doxorubicin and hypoxia [144]. This was achieved by inhibiting the Bcl2-interacting protein 3 (Bnip3) from targeting the mitochondria. Although ellagic acid does not have any effect on mRNA level of Bnip3, it completely prevented mitochondrial disturbances, including mPTP opening, loss of mitochondrial membrane potential and mitochondrial fission induced by doxorubicin and hypoxia. This shows that ellagic acid has no effect on the transcription of the Bnip3 promoter or gene but is likely to exert a post-translational inhibiting effect on the Bnip3 protein instead.

Moreover, Dhingra et al. [145] also reported that ellagic acid was able to inhibit doxorubicin-induced mitophagy, also by antagonizing mitochondrial targeting of Bnip3, which is the primary underlying event for mitochondrial fragmentation and mitophagy in cardiomyocytes. Ellagic acid also suppressed lactate dehydrogenase (LDH) release and loss of the high mobility group protein B1 (HMGB1), which are both necrotic cell death markers [145]. This highlights the association between mitochondrial injury and necrotic cell death in cardiomyocytes.

Khanlou et al. [146] also reported that ellagic acid protects mitochondria against toxicity induced by bevacizumab, an anthracycline agent, by either its antioxidant properties or indirectly via maintenance of mitochondrial complex II activity [147].

### 6.3. Gallic Acid

Gallic acid, or 3,4,5 trihydroxybenzoic acid, is commonly found in many plants and is a second metabolite [148]. Gallic acid prevents mitochondria-originated oxidative stress and protects against DNA damage and apoptosis of cells [149]. The effect of gallic acid, in its native form, has not been studied in detail on mitochondrial respiration in animal cells, but its esters have been reported to affect the mitochondrial bioenergetics. With all the compounds, the inhibition was diagnosed with the complex I electron carrier function (NADH-CoQ), preventing the generation of the proton gradient and ATP synthesis, eventually leading to cell death [150].

Although a detailed study on the effect of gallic acid in DCM is not available, gallic acid in synergy with cyclosporine protects myocardial infarction and associated necrotic cell death in a rat model of myocardial damage. The histopathological observations indicate that the cellular morphologies in the affected cardiac tissues were restored to normal in the treatment group where gallic acid provides antioxidant defense, along with cyclosporine, the mPTP inhibitor [151].

Methyl gallate, a methyl ester gallic acid, exhibits a better cardioprotective effect in comparison to gallic acid, as observed in neonatal rat cardiomyocytes challenged with hydrogen peroxide, and it improves mitochondrial functions and increases glutathione synthesis [152]. Epigallocatechin gallate, also another gallic acid ester, protects cardiomyocytes against fluoride-induced toxicity by improving the functions of several mitochondrial enzymes involved in respiration and metabolism, consequently down-regulating oxidative stress-induced apoptosis [153]. It also protects against autophagic cell death, which is usually developed in diabetic cells with mitochondrial dysfunctions [154].

### 6.4. Salvianolic Acid

Salvianolic acids, especially salvianolic acid A (Sal A) and salvianolic acid B (Sal B), have been found to have potent antioxidative capabilities due to their polyphenolic structure [155]. Indeed, both Sal B and Sal A show a high radical scavenging capacity, which can be measured by neutralizing free radical assays, such as the DPPH radical scavenging test or the 2,2′-Azinobis-(3-Ethylbenzthiazolin-6-Sulfonic Acid (ABTS) assay [156,157]. It was found that Sal A was the most potent antioxidant among the salvianolic acids. Sal B exhibited higher scavenging activities than vitamin C against hydroxyl radical (HO), superoxide radical (O_2_^−^), 2,2-diphenyl-1-picryl-hydrazyl-hydrate (DPPH) radicals, and ABTS radicals. However, their iron chelating and H_2_O_2_ scavenging activities were lower than vitamin C [156]. Liu et al. [158] reported that Sal B prevents mitochondrial dysfunction by scavenging ROS. Pre-treatment with Sal B in hepatocytes prohibits mitochondrial fragmentation by decreasing mitochondrial fission under oxidative stress conditions. Sal B also prevents H_2_O_2_-induced oxidative stress by regulating mitochondrial morphology and function. Intravenous administration of Sal A (0.3–3 mg/kg) significantly attenuated isoproterenol-induced cardiac dysfunction and myocardial injury, and improved mitochondrial respiratory functions in a rat model with isoproterenol-induced myocardial infarction [159].

Salvianolic acids have been reported to protect cardiomyocytes from drug-induced toxicity due to their ROS scavenging ability. It was noted that Sal A converted HO generated by electron transfer from adriamycin semiquinone radicals to H_2_O_2_ in adriamycin-induced mitochondrial toxicity of a rat heart in a dose-dependent manner [160]. In mice with doxorubicin-induced cardiotoxicity, salvianolic acids (containing 64.92% Sal B, 40 mg/kg/day for 3 days) also protected myocardium by reducing oxidative stress [161].

### 6.5. Chlorogenic Acid

Chlorogenic acid, a well-known caffeoylquinic acid from natural herbs, has been shown to exert antioxidant and multiple bioactivities. Chlorogenic acid is able to protect against degenerative and age-related diseases in animals and contributes to the prevention of cardiovascular diseases [162]. Akila et al. [163] reported that oral administration of chlorogenic acid decreased lysosomal enzyme activities of the hearts of isoproterenol-induced rats. The improved stability of lysosome might be due to this compound action in reducing membrane damage, thus inhibiting the release of the lysosomal enzyme. Isoproterenol also damages the heart by increasing ROS production, reducing antioxidative capacity and altering the mitochondrial respiratory chain function via down-regulation of specific mitochondrial respiratory enzyme activities [164]. However, pre-treatment of chlorogenic acid was found to restore the activities of these enzymes, which were affected due to disturbance in mitochondrial substrate oxidation [163]. Li et al. [165] also reported that chlorogenic acid was able to reduce mitochondrial ROS production following myocardial infarction.

### 6.6. Rosmarinic Acid

Rosmarinic acid is extracted from plants containing polyphenol hydroxyl acid with a structure of ester of caffeic acid and 3-(3,4-dihydroxyphenyl) lactic acid [166]. Rosmarinic acid reduces intracellular ROS production of H9c2 cells by suppressing the opening of the mPTP and inhibition of apoptotic factors from mitochondria [167]. Activation of the mPTP induces depolarization of the mitochondrial membrane, leading to the mitochondrial release of apoptotic proteins such as cytochrome C. This eventually leads to the activation of caspase-9 and caspase-3, forming apoptotic bodies [168].

Rosmarinic acid was also able to protect cardiac cells against doxorubicin-induced damage by suppressing energy deprivation and ATP degradation [169]. Rosmarinic acid was also found to be more effective in protecting cardiomyocytes compared to chlorogenic acid and caffeic acid, which might be attributable to the number of hydroxyl constituents in the compounds.

### 6.7. Vanillic Acid

Vanillic acid is found in secondary plant products and is widely used as a flavoring agent, a food additive and a preservative [170]. It was found that vanillic acid possesses potent antioxidative, antihypotensive, and antiapoptotic properties, as well as being cardioprotective [170,171,172,173]. Vanillic acid has been widely studied as a cardioprotective agent, and most reports relate its protective effect on its properties as a potent antioxidant. Baniahmad et al. [174] showed that vanillic acid decreased cardiotoxicity biomarkers and oxidative stress and suppressed TLR4 signaling, consequently disrupting inflammation pathways.

Yao et al. [175] reported that vanillic acid protects H9c2 cardiomyoblast cells against hypoxia/re-oxygenation injury by reducing ROS generation, stabilizing mitochondrial membrane potential, limiting the mPTP opening, decreasing caspase-3 activity and inhibiting cardiomyocyte apoptosis. Interestingly, upon knocking out AMPKα2-siRNA in H9c2 cells, the antioxidative effects of vanillic acid vanished. This finding confirmed that vanillic acid exerts its cardioprotective effects via the AMPK signaling pathway.

### 6.8. Caffeic Acid

Caffeic acid (3,4-dihydroxycinnamic acid) is one of the most common phenolic acids that occur in fruits, grains, and dietary supplements [176]. It has been previously reported that caffeic acid possesses a cardioprotective property via its free radical-scavenging effects [177,178].

The enhanced activities of tricarboxylic cycle enzymes in caffeic acid-pretreated isoproterenol-induced rats could be due to the ability of caffeic acid to scavenge ROS. Kumaran et al. [179] reported that pre-treatment of caffeic acid in rats reduces cytochrome-C-oxidase and NADH dehydrogenase, demonstrating that caffeic acid is able to prevent membrane phospholipid degradation. Mitochondrial Ca^2+^ levels were also reduced with caffeic acid pre-treatment, while ATP levels increased significantly. The tricarboxylic cycle of the mitochondria was observed to be increased in addition to the increase in respiratory chain enzyme activity in the hearts of isoproterenol-induced rats. This results in the increased production of ATP. In short, caffeic acid is shown to maintain a normal mitochondrial function and structure, thus protecting the cardiac mitochondrial against isoproterenol-induced perturbation. In addition, superoxide radicals formed on complex III of cardiac mitochondrial respiratory chain was also found to be effectively scavenged by caffeic acid treatment [180].

Levels of mitochondrial lipids were also restored by caffeic acid pretreatment, showing that this phenolic acid aids in maintaining the integrity and stability of the mitochondrial membranes and prepares them against isoproterenol [179]. In addition, caffeic acid was also shown to protect mitochondria against structural alterations.

## 7. Conclusions

In summary, phenolic acids were found to protect cardiac mitochondria against perturbations such as the mPTP opening, disruption in the mitochondrial membrane potential and excessive mitochondrial ROS production via different pathways. The protective effects also include direct action of these phenolic acids as potent antioxidants, providing aid in reducing both cytosolic and mitochondrial oxidative stress. However, it is important to highlight that a huge portion of the studies of phenolic acid effects on cardiac mitochondrial are mostly focused on mitochondrial dysfunction in the ischemic/hypoxic condition as well as post-myocardial infarction, while only a handful focus on DCM models. As mitochondrial dysfunction-induced damage in DCM involves various pathways, treatment of these phenolic acids might have different effects. Phenolic acids are proven to protect the heart against mitochondrial dysfunction (Figure 3), thus prompting further investigation for their effects on mitochondrial dysfunction in DCM. The fact that phenolic acids are obtainable from natural sources and are evidently cardioprotective implies their potential to be used in diets due to their nutritional effects.

Due to the increase in the prevalence of diabetes mellitus globally, we may predict that our societies will have an impulsive increase in DCM. As for future direction, more extended studies will be required, and it is important to add to the current studies to completely elucidate the molecular basis of mitochondrial dysfunction in DCM. More novel yet relatively under-explored hypotheses may be integrated to improve our fundamental understanding of how phenolic acids play roles and could possibly be used as therapeutic treatments for DCM by targeting mitochondrial dysfunction. Thus, additional mechanistic studies are warranted, using both in vivo and in vitro assays that address fundamental underlying pathophysiological mechanisms with the goal to reveal useful targets for the development of therapeutic strategies (clinical studies) using phenolic acids that may prevent DCM in diabetic patients.

## Figures and Tables

**Figure 1 ijms-21-06043-f001:**
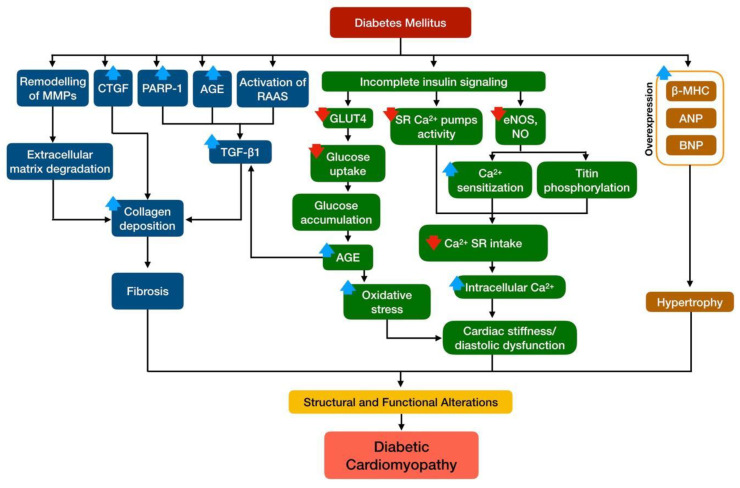
Overview of structural and functional alterations of diabetic cardiomyopathy.

**Figure 2 ijms-21-06043-f002:**
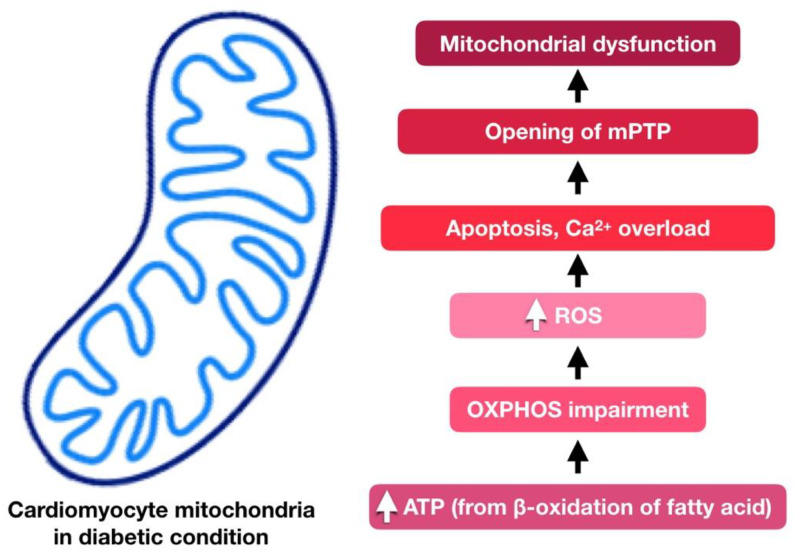
Involvement of mitochondrial dysfunction in diabetic cardiomyopathy. The adenosine triphosphate (ATP) was generated from fatty acid oxidation instead of glucose in the diabetic condition, which generates more ROS and disrupts the oxidative phosphorylation process. Consequently, apoptosis occurs following impairment of mitochondrial Ca^2+^ handling, which leads to mitochondrial respiratory dysfunction. Ca^2+^ overload also causes mitochondria permeability transition pore (mPTP) opening, leading to mitochondrial dysfunction.

**Figure 3 ijms-21-06043-f003:**
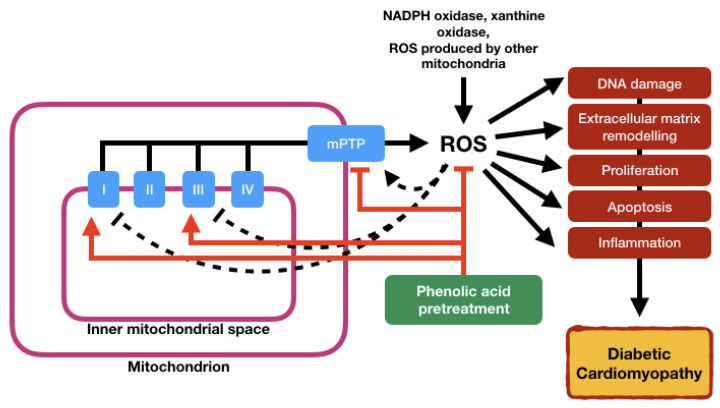
Mitochondrial dysfunction in diabetic cardiomyopathy and protection of phenolic acids. An initial burst of ROS (e.g., from NADPH oxidase or other mitochondria) leads to the opening of the mPTP and depolarization of the mitochondrial membrane potential. Due to increased electron flux along the ETC, mitochondrial ROS production will increase (ROS-induced ROS release) and may enter the cytosol via the opened mPTP. Inhibition of Complex I and II of the mitochondria leads to further formation of ROS. Mitochondrial ROS formation can disturb the function of cardiomyocytes in several ways, including DNA damage, external matrix remodeling, proliferation, inflammation or apoptosis. These molecular changes build the basis for the development of myocardial injury and eventually heart failure. In contrast, preconditioning with phenolic acid prevents myocardial injury by preventing mPTP from opening, resulting in decreased mitochondrial ROS production. Antioxidative action of the phenolic acid represents another strategy to prevent oxidative damage of cardiomyocytes.

**Table 1 ijms-21-06043-t001:** Summary of *in vivo* and *in vitro* treatments of phenolic acid on cardiac mitochondrial dysfunction.

Phenolic Acid	Dose/Concentration	Results	References
Ferulic acid	*In vitro*: 5.5 μM	Improves expression of mitochondrial biogenesis and dynamics markers and reduces oxidative stress.	Perez-Ternero et al., 2017
	*In vitro*: 10 and 25 μM	Protects H9c2 cardiomyoblasts from hyperglycemia-induced oxidative stress damages via maintenance of Ca^2+^ homeostasis by safeguarding the SERCA/PLN pathway and mitochondrial function.	Salin Raj et al., 2019
	*In vitro*: 1, 5, 10 µg/mL	Mitigated oxidative damage via antioxidative protection by heme oxygenase-1 upregulation.	Song et al., 2016
Ellagic acid	*In vitro*: 1–20 μM	Suppress oxidative mitochondrial injury induced by doxorubicin and hypoxia via inhibition of Bnip3. Prevents cell necrosis by suppressing LDH release and loss of high mobility group protein B1.	Dhingra et al., 2017
	*In vitro*: 10–100 µM	Protects mitochondria against toxicity induced by bevacizumab, an anthracycline agent, by either its antioxidant properties or indirectly via maintenance of mitochondrial complex II activity.	Khanlou et al., 2020
Gallic acid	*In vitro*: 50 μM	Methyl gallate improves mitochondrial functions and increases glutathione synthesis.	Khurana et al., 2014
	*In vivo*: 7.5, 15, 30 mg/kg/day	Gallic acid in synergy with cyclosporine protects myocardial infarction and associated necrotic cell death.	Dianat et al., 2014
	*In vivo*: 40 mg/kg/day	Epigallocatechin gallate, a gallic acid ester, protects cardiomyocytes by improving the functions of several mitochondrial enzymes involved in respiration and metabolism, consequently down-regulating the oxidative stress-induced apoptosis.	Miltonprabu and Thangapandiyan 2015
Salvianolic acid	*In vitro*: 10 mM	Sal B prevents mitochondrial dysfunction by scavenging ROS, prohibits mitochondrial fragmentation by decreasing mitochondrial fission under oxidative stress conditions.	Liu et al., 2017
	*In vivo*: 0.3–3 mg/kg	Sal A attenuates isoproterenol-induced cardiac dysfunction and myocardial injury and improves mitochondrial respiratory function.	Wang et al., 2009
	*In vivo*: 40 mg/kg/day	Sal B protects myocardium through reducing oxidative stress.	Jiang et al., 2008
Chlorogenic acid	*In vivo*: 40 mg/kg	Improved lysosomal stability of the heart of isoproterenol-induced rats.	Akila et al., 2017
	*In vivo*: 20 mg/kg	Reduce mitochondrial ROS production following myocardial infarction.	Li et al., 2018
Rosmarinic acid	*In vitro*: 5, 20, 50 mM	Reduces intracellular ROS production of H9c2 cells by suppressing the opening of the mPTP and inhibition of apoptotic factors from mitochondria.	Diao et al., 2016
	*In vitro*: 100, 200 µm	Protect cardiac cells against doxorubicin-induced damage by suppressing energy deprivation and ATP degradation.	Chlopcíková et al., 2004
Vanilic acid	*In vivo*: 10, 20, and 40 mg/kg	Suppressed Toll-like receptor 4 signaling, consequently disrupting the inflammation pathway.	Baniahmad et al., 2020
	*In vitro*: 1.00 mM	Reducing of ROS generation, stabilizing mitochondrial membrane potential, limiting the mPTP opening, decreasing caspase-3 activity and inhibiting cardiomyocyte apoptosis. Exerts its cardioprotective effects via the AMPK signaling pathway.	Yao et al., 2020
Caffeic acid	*Ex vivo*: 1 μM to 1 mM	Caffeic acid scavenges mitochondria-produced superoxides.	Dudylina et al., 2018
	*In vivo*: 15 mg/kg	Pre-treatment of caffeic acid in rats reduces cytochrome-C-oxidase and NADH dehydrogenase, demonstrating that caffeic acid is able to prevent membrane phospholipid degradation.	Kumaran and Prince 2010

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
