# Peer review of "Mitochondrial Dysfunction in Diabetic Cardiomyopathy: The Possible Therapeutic Roles of Phenolic Acids"

_ijms, 2020, doi:10.3390/ijms21176043_

Round 1

Reviewer 1 Report

In this review, the authors summarise the role of mitochondrial dysfunction in diabetes-related cardiomyopathy and explore the role of different phenolic acid derivatives in the management of this condition. The area is topical and although there are several published reviews to date, the role of the phenolic acids is yet to be comprehensively discussed. I have the following comments:

  1. It would be helpful if the authors acknowledge the different published reviews early on and clarify the novel aspects of their work, in order to keep the reader engaged.
  2. The authors did not mention their search strategy and this needs to be clarified.
  3. In line 36, it is mentioned that “DCM is typically presented with a marked increase of HDL…”. To my knowledge, this is incorrect as HDL is not raised in DCM; can the authors look into this?
  4. Section 2, line 46: this does not appear to flow well and tends to jump between sections. For example, paragraph 2 is about fibrosis, paragraph 3 is molecular mechanisms then paragraph 4 is about fibrosis again. I kindly ask the authors to rethink the structure of this section with special emphasis on molecular mechanisms (which can be expanded), given the interests of the journal. A Figure summarising molecular mechanisms would be helpful.
  5. Section 4 would benefit from a Figure summarising the role of mitochondrial dysfunction in DCM.
  6. While the authors successfully make the link between oxidative stress and mitochondrial dysfunction, this is somewhat less clear with inflammation, a section that would benefit from making it more focused.
  7. The two main metabolic abnormalities in diabetes, namely hyperglycaemia and insulin resistance (IR), are not always well discussed in relation to mitochondrial dysfunction. The authors may wish to include a section to clearly discuss the molecular mechanisms by which high glucose levels and IR cause mitochondrial dysfunction. Also, It would be helpful to add a paragraph describing the main differences between type 1 and type 2 diabetes in relation to DCM.
  8. A Table summarising the phenolic acids discussed (mode of action, sources, main effects on mitochondria and cardiac function in animal/and or human studies) would be of great help to the reader. Also, when discussing various studies with phenolic acids, the distinction between in vitro, and in vivo animal and human studies should be made clear.
  9. The conclusion section can be expanded slightly (few more sentences) to include future directions and views of the authors on the studies needed in this area.
  10. Minor points:
    1. As the authors mention, studies used cardiac ultrasound and Magnetic resonance imaging to assess DCM. However, the two techniques have very different sensitivities, which often lead to different conclusions and this should be made clear to the reader.
    2. Line 101: replace “Mitochondrial” with “Mitochondria”.
    3. Figure legend should better explain what I, II, III and IV mean in the figure with clarification of all the abbreviations used. Also, the figure would benefit from having diabetes-specific pathways added.
    4. The term diabetic patients is not preferred and best to replace with "individuals with diabetes" or "patients with diabetes". Diabetic cardiomyopathy is fine.

Author Response

Ijms-858503

Response to reviewers

Thank you for giving us the opportunity to submit a revised draft of the manuscript “Mitochondrial Dysfunction in Diabetic Cardiomyopathy: Possible Therapeutic Roles of Phenolic Acids” for publication in the International Journal of Molecular Science. We appreciate the time and effort that  reviewers dedicated for providing feedback on our manuscript and are grateful for the insightful comments on and valuable improvement to our paper. We have incorporated most of the suggestions made by the reviewers. Those changes are highlighted within the manuscript. Please see below, in blue, for a point-by-point response to the reviewers’ comments and concerns. All page numbers refer to the revised manuscript file with tracked changes.

Comments and Suggestions for Authors

Reviewer #1

In this review, the authors summarise the role of mitochondrial dysfunction in diabetes-related cardiomyopathy and explore the role of different phenolic acid derivatives in the management of this condition. The area is topical and although there are several published reviews to date, the role of the phenolic acids is yet to be comprehensively discussed.

Author response: We would like to thank Reviewer 1 for the constructive suggestions and comments. We appreciate it as it has made our review manuscript.

I have the following comments:

  1. It would be helpful if the authors acknowledge the different published reviews early on and clarify the novel aspects of their work, in order to keep the reader engaged.

Author response: Thank you to the reviewer for the suggestion.  We have included a paragraph [Page 2, Line 65] on different published reviews and clarified the novel aspects of their work.

  1. The authors did not mention their search strategy and this needs to be clarified.

Author response: Thanks to the reviewer for pointing this out.  We have included the search strategy in the manuscript accordingly [Page 2, Line 82].

  1. In line 36, it is mentioned that “DCM is typically presented with a marked increase of HDL…”. To my knowledge, this is incorrect as HDL is not raised in DCM; can the authors look into this?

Author response: Thank you for pointing out the error. Indeed, among clinical features in DCM is increased levels of LDL and decreased level of HDL. We have amended this mistake in the manuscript [Page 2, Line 98].

  1. Section 2, line 46: this does not appear to flow well and tends to jump between sections. For example, paragraph 2 is about fibrosis, paragraph 3 is molecular mechanisms then paragraph 4 is about fibrosis again. I kindly ask the authors to rethink the structure of this section with special emphasis on molecular mechanisms (which can be expanded), given the interests of the journal. A Figure summarising molecular mechanisms would be helpful.

Author response: Thank you for your opinion and suggestions. As suggested, we had restructure the whole section [Page 3 Line 112] and added a figure (Figure 1) to briefly summarise the molecular mechanism in the development of DM.

  1. Section 4 would benefit from a Figure summarising the role of mitochondrial dysfunction in DCM.

Author response: As suggested by the reviewer, a figure (Figure 2) summarizing role of mitochondrial dysfunction in DCM section 4 has been included in the manuscript to assist understanding.

  1. While the authors successfully make the link between oxidative stress and mitochondrial dysfunction, this is somewhat less clear with inflammation, a section that would benefit from making it more focused.

Author response: Thank you reviewer for the opinions. We had restructure the section and focused more on the mitochondrial dysfunction-derived cardiac inflammation in diabetic condition to the roles of NLRP3 inflammasome [Page 6, Line 264].

  1. The two main metabolic abnormalities in diabetes, namely hyperglycaemia and insulin resistance (IR), are not always well discussed in relation to mitochondrial dysfunction. The authors may wish to include a section to clearly discuss the molecular mechanisms by which high glucose levels and IR cause mitochondrial dysfunction. Also, it would be helpful to add a paragraph describing the main differences between type 1 and type 2 diabetes in relation to DCM.

Author response: Based on the suggestion given, we added another section to discuss how mitochondrial dysfunction causes insulin resistance that further aggravate the development and condition of DCM [Page 8, Line 359]. Additionally, we had also added the comparison of type 1 and type 2 diabetes in relation to DCM in the Introduction part.

  1. A Table summarising the phenolic acids discussed (mode of action, sources, main effects on mitochondria and cardiac function in animal/and or human studies) would be of great help to the reader. Also, when discussing various studies with phenolic acids, the distinction between in vitro, and in vivo animal and human studies should be made clear.

Author response: We had added a table to summarise each phenolic effect based on the type of study, the dose and effects on cardiac mitochondria. [Page 12].

  1. The conclusion section can be expanded slightly (few more sentences) to include future directions and views of the authors on the studies needed in this area.

Author response: As suggested, we have expanded the conclusion section and added future direction and our view on the studies [Page 14, Line 592].

  1. Minor points:
    1. As the authors mention, studies used cardiac ultrasound and Magnetic resonance imaging to assess DCM. However, the two techniques have very different sensitivities, which often lead to different conclusions and this should be made clear to the reader.

Author response: Thank you for your suggestions. We had added additional explanation on these diagnostic techniques as suggested. [Page 3 Line 118]

Both echocardiography and MRI are equally sensitive in detecting structural abnormalities on the heart. However, MRI has the capability to detect more pathological abnormalities. Considering the cost of running an MRI and how it is not as widely available as echocardiography, with comparable sensitivity, echocardiography is still a reliable diagnostic tool for early detection of DCM (Khalil & Alzahrani 2020).”

  1. Line 101: replace “Mitochondrial” with “Mitochondria”.

Author response: “Mitochondrial” has been replaced with “Mitochondria” (Highlighted throughout the text)

  1. Figure legend should better explain what I, II, III and IV mean in the figure with clarification of all the abbreviations used. Also, the figure would benefit from having diabetes-specific pathways added.

Author response: Thank you for your comment. We had amended the figure legend with additional information.

  1. The term diabetic patients is not preferred and best to replace with "individuals with diabetes" or "patients with diabetes". Diabetic cardiomyopathy is fine.

Author response: The term has been corrected accordingly and are highlighted throughout the manuscript.

Thank you very much for your time. We truly appreciate it.

Best regards,

Fatin Farhana Jubaidi

Reviewer 2 Report

The authors provide a relatively comprehensive review of mitochondrial dysfunction and that in DCM and follow up with a reasonable review of phenolic acids and their effects in limiting mitochondrial dysfunction and providing a protective effect and inferring therapeutic potential. Within the length of this review, the scope and detail provided gives important insights into mechanisms of mitochondrial dysfunction and approaches to limit or prevent dysfunction. The same is true for the phenolic acid discussion.

One aspect of the discussion that could be expanded to provide better insights is how phenolic acids can be studied effectively and how they can be utilized effectively, either pharmacologically or through modified diet in DCM. This would help to focus mitochondrial and phenolic acid backgrounds on DCM.

One issue of accuracy,

Line 93:  isovolumic relaxation should be “isovolumic relaxation time”

Author Response

Ijms-858503

Response to reviewers

Dear Dr Ling Zhu,

Thank you for giving us the opportunity to submit a revised draft of the manuscript “Mitochondrial Dysfunction in Diabetic Cardiomyopathy: Possible Therapeutic Roles of Phenolic Acids” for publication in the International Journal of Molecular Science. We appreciate the time and effort that you and the reviewers dedicated for providing feedback on our manuscript and are grateful for the insightful comments on and valuable improvement to our paper. We have incorporated most of the suggestions made by the reviewers. Those changes are highlighted within the manuscript. Please see below, in blue, for a point-by-point response to the reviewers’ comments and concerns. All page numbers refer to the revised manuscript file with tracked changes.

Comments and Suggestions for Authors

Reviewer #2

The authors provide a relatively comprehensive review of mitochondrial dysfunction and that in DCM and follow up with a reasonable review of phenolic acids and their effects in limiting mitochondrial dysfunction and providing a protective effect and inferring therapeutic potential. Within the length of this review, the scope and detail provided gives important insights into mechanisms of mitochondrial dysfunction and approaches to limit or prevent dysfunction. The same is true for the phenolic acid discussion.

Author response: We would like to thank Reviewer 2 for the positive comments. We appreciate your time.

One aspect of the discussion that could be expanded to provide better insights is how phenolic acids can be studied effectively and how they can be utilized effectively, either pharmacologically or through modified diet in DCM. This would help to focus mitochondrial and phenolic acid backgrounds on DCM.

Author response: Thanks for the suggestion.  We have included these comments (also as suggested by review 1) accordingly [Page 14, Line 588] in the conclusion part. A summary table including in vivo and in vitro studies have been included for better understanding too [Page 12].

One issue of accuracy,

Line 93:  isovolumic relaxation should be “isovolumic relaxation time”

Author response: “isovolumic relaxation” has been replaced with “isovolumic relaxation time” (line 127)

Thank you very much for your time. We truly appreciate it.

Best regards,

Fatin Farhana Jubaidi